# Congenital Oropouche in Humans: Clinical Characterization of a Possible New Teratogenic Syndrome

**DOI:** 10.3390/v17030397

**Published:** 2025-03-11

**Authors:** Bethânia de Freitas Rodrigues Ribeiro, André Rodrigues Façanha Barreto, André Pessoa, Raimunda do Socorro da Silva Azevedo, Flávia de Freitas Rodrigues, Bruna da Cruz Beyruth Borges, Natália Pimentel Moreno Mantilla, Davi Dantas Muniz, Jannifer Oliveira Chiang, Lucas Rosa Fraga, Fernanda Sales Luiz Vianna, Maria Teresa Vieira Sanseverino, Lilith Schuler Faccini, Fernanda Eduarda das Neves Martins, Rafael da Silva Azevedo, Lívia Carício Martins, Livia Medeiros Neves Casseb, Consuelo Silva Oliveira, Pedro Fernando da Costa Vasconcelos, Juarez Antônio Simões Quaresma, Alberto Mantovani Abeche, Vania de Mesquita Gadelha Prazeres, Lucia Andreia Nunes de Oliveira, Simone de Menezes Karam, Giulia Radin, Miguel Del Campo, Camila V. Ventura, Lavinia Schuler-Faccini

**Affiliations:** 1Secretaria de Estado de Saúde do Acre (SESACRE), Rio Branco 69900-376, AC, Brazil; bfrodrigues@gmail.com (B.d.F.R.R.); rubeyruth@hotmail.com (B.d.C.B.B.); natmorenomantilla@gmail.com (N.P.M.M.); 2Hospital Universitário Walter Cantídio (HUWC), Radiology Department and Universidade Federal do Ceará (UFC), Fortaleza 60020-181, CE, Brazil; andrefacanha@yahoo.com.br; 3Hospital Infantil Albert Sabin, Universidade Federal do Ceará (UFC), Fortaleza 60020-181, CE, Brazil; andrepessoa10@yahoo.com.br; 4Section on Arbovirology and Hemorrhagic Fever, Instituto Evandro Chagas/SVSA/MS, Ananindeua 67030-000, PA, Brazil; raimundaazevedo@iec.gov.br (R.d.S.d.S.A.); janniferchiang@iec.gov.br (J.O.C.); liviamartins@iec.gov.br (L.C.M.); liviacasseb@iec.gov.br (L.M.N.C.); consuelooliveira@iec.gov.br (C.S.O.); pedro.vasconcelos@uepa.br (P.F.d.C.V.); 5Hospital Universitário de Brasília (UnB), Brasília 70330-750, DF, Brazil; flavinhafrodrigues@gmail.com; 6Departamento de Neurologia e Neurofisiologia, Universidade Federal de São Paulo (Unifesp), São Paulo 04021-001, SP, Brazil; ddmuniz@huhsp.org.br; 7Department of Morphological Sciences, Universidade Federal do Rio Grande do Sul (UFRGS), Porto Alegre 90040-060, RS, Brazil; lrfraga@hcpa.edu.br; 8Brazilian Teratogen Information System, SIAT, Hospital de Clinicas de Porto Alegre, Universidade Federal do Rio Grande do Sul, Porto Alegre 90035-903, RS, Brazil; fvianna@hcpa.edu.br (F.S.L.V.); msanseverino@hcpa.edu.br (M.T.V.S.); aabeche@hcpa.edu.br (A.M.A.); radingiulia@gmail.com (G.R.); 9Graduate Program in Health Sciences: Medicine, Universidade Federal do Rio Grande do Sul, Porto Alegre 90040-060, RS, Brazil; lsfaccini@hcpa.edu.br; 10Graduate-Program in Genetics and Molecular Biology, Universidade Federal do Rio Grande do Sul (UFRGS), Porto Alegre 90040-060, RS, Brazil; vaniaprazeres@ufam.edu.br; 11Department of Pediatrics, School of Medicine, Pontificia Universidade Católica do Rio Grande do Sul (PUCRS), Porto Alegre 90619-900, RS, Brazil; 12Postgraduate Program in Virology, Instituto Evandro Chagas/SVSA/MS, Ananindeua 67030-000, PA, Brazil; fernandamartins@iec.gov.br; 13Hospital Geral de Belém, Belém 66050-450, PA, Brazil; r.azevedo4556@gmail.com; 14Department of Patology, Universidade do Estado do Pará, Belém 66050-540, PA, Brazil; 15Tropical Medicine Center, Universidade Federal do Pará, Belém 66050-540, PA, Brazil; juarez@ufpa.br; 16Department of Ginecology and Obstetrics, Hospital de Clínicas de Porto Alegre, Universidade Federal do Rio Grande do Sul, Porto Alegre 90040-060, RS, Brazil; 17Department of Maternal and Fetal Health, School of Medicine, Universidade Federal do Amazonas, Manaus 69020-160, AM, Brazil; 18Graduate Program in Child and Adolescent Health, Universidade Federal do Rio Grande do Sul (UFRGS), Porto Alegre 90040-060, RS, Brazil; lucia.oliveira@ufrgs.br; 19Department of Pediatrics, School of Medicine, Universidade Federal de Rio Grande (FURG), Pelotas 96090-790, RS, Brazil; karam.simone@gmail.com; 20Department of Pediatrics, University of California San Diego, La Jolla, CA 92093, USA; 21Department of Ophthalmology, Altino Ventura Foundation (FAV), Recife 50731-490, PE, Brazil; camilaventuramd@gmail.com; 22Medical Genetics Service—Hospital de Clínicas de Porto Alegre (HCPA), Universidade Federal do Rio Grande do Sul (UFRGS), Av. Ramiro Barcelos, 2350, Porto Alegre 91035-903, RS, Brazil

**Keywords:** Oropouche fever, vertical transmission, congenital anomalies, microcephaly

## Abstract

Oropouche fever is caused by the Oropouche virus (OROV; Bunyaviridae, Orthobunyavirus), one of the most frequent arboviruses that infect humans in the Brazilian Amazon. This year, an OROV outbreak was identified in Brazil, and its vertical transmission was reported, which was associated with fetal death and microcephaly. We describe the clinical manifestations identified in three cases of congenital OROV infection with confirmed serology (OROV-IgM) in the mother-newborn binomial. One of the newborns died, and post-mortem molecular analysis using real-time RT-qPCR identified the OROV genome in several tissues. All three newborns were born in the Amazon region in Brazil, and the mothers reported fever, rash, headache, myalgia, and/or retro-orbital pain during pregnancy. The newborns presented with severe microcephaly secondary to brain damage and arthrogryposis, suggestive of an embryo/fetal disruptive process at birth. Brain and spinal images identified overlapping sutures, cerebral atrophy, brain cysts, thinning of the spinal cord, corpus callosum, and posterior fossa abnormalities. Fundoscopic findings included macular chorioretinal scars, focal pigment mottling, and vascular attenuation. The clinical presentation of vertical OROV infection resembled congenital Zika syndrome to some extent but presents some distinctive features on brain imaging and in several aspects of its neurological presentation. A recognizable syndrome with severe brain damage, neurological alterations, arthrogryposis, and fundoscopic abnormalities can be associated with in utero OROV infection.

## 1. Introduction

The Oropouche virus (*Orthobunyavirus oropoucheense*; OROV) is an arthropod-borne virus first isolated in 1955 [1]. OROV is transmitted by midge bites, especially *Culicoides paraensis*, an anthropophilic fly largely distributed in the Americas. Since the 1960s, there have been reports of outbreaks in human populations in the Brazilian Amazon region and other South American countries. Recently, OROV outbreaks have spread to all regions of Brazil. Exported cases were detected in Europe and North America [2].

Oropouche infection is characterized by fever, headache, myalgia, arthralgia, nausea, retro-orbital pain, and cutaneous rash. Neurological complications such as encephalitis and meningitis could arise [3].

In June 2024, the Brazilian Ministry of Health (MoH) released a technical note alerting of the possible vertical transmission of the OROV in one fetal death at 30 weeks of pregnancy. In this case, OROV RNA was detected in cord blood and placenta, and the fetal organs were tested [4]. In July, the Pan American Health Organization also published an epidemiological alert on the vertical transmission of the OROV in the region of the Americas [5]. In October, six cases of babies born from 2015 to 2024 in Brazil with microcephaly and other congenital abnormalities with laboratory-positive results for OROV were reported [6].

The three cases born in 2024 were clinically evaluated by our team. Here, we present a detailed clinical picture of this possibly new teratogenic syndrome.

## 2. Materials and Methods

The cases included herein were born in the Barbara Heliodora Maternity in Rio Branco, Acre, Brazil, and received all the care needed for their condition. including the publication of photographs of the full face. This is part of a project for surveillance of congenital anomalies in Brazil, approved by the ethics committee and registered under the protocol CAAE 67379223.7.1001.532.

IgM antibodies in serum and CSF were tested using the IgM Capture Immunoenzymatic Assay (IgM-ELISA). Optical density (OD) ≥ 0.301 suggests a recent infection, with a sensitivity of 93% and specificity of 99%. The detection of total antibodies was performed through the Hemagglutination Inhibition (HI) test. Reactive results (dilution titers ≥ 1:20) indicate previous contact with OROV. RNA was extracted from serum and CSF with the QIAamp^®^ Viral RNA Mini kit (Qiagen, Hilden, Germany), and for tissue fragments, the TRIzol Plus RNA Purification kit (Invitrogen, Waltham, MA, USA). To detect OROV RNA, RT-qPCR was performed using the commercial kit Superscript III Platinum One-Step RT-qPCR System^®^ with separate Rox (Invitrogen). The OROV-specific primers and probe were previously described [7]. RT-qPCR reactions were carried out in duplicate. Samples were considered detectable for OROV RNA when cycle threshold values were lower than 38 in both replicates.

## 3. Case Reports

A detailed clinical description is presented in Table 1, and Table 2 presents detailed laboratory results from both mothers and the children.

### 3.1. Case 1 (Figure 1)

Case 1 is a female infant born at 38 weeks gestational age in April 2024. At birth, her length was 45 cm (−1.8 Z), weight was 2740 g (−0.75 Z), and head circumference was 30 cm (−2.66 Z). The mother reported fever, rash, headache, myalgia, and retro-orbital pain during the 21st week of pregnancy. Serological tests were positive for OROV-IgM, and other infections were ruled out (Table 1). In the 33rd week of pregnancy, an ultrasound detected a small fetus with microcephaly compared to its gestational age, ventriculomegaly, porencephalic cysts, and absence of the corpus callosum. The physical exam after birth (Table 1) revealed craniofacial disproportion with a small skull, uneven overlapping cranial sutures, and posterior prominence of the occipital bone with loose and redundant skin. In the first months of life, the patient presented with decreased mobility in the limbs, with hyperreflexia and axial/appendicular hypotonia. In the neurological examination performed at 6 months of age, the pyramidal signs (strength deficit and hyperreflexia) remained associated with the persistence of the tonic-cervical reflex and appendicular dystonia. Serological testing collected on the fifth day of life was positive for IgM-OROV. Other congenital or arboviral infections, exposure to teratogens, and known genetic disorders were ruled out (Table 1 and Table 2).

Transfontanellar ultrasound showed cystic leukomalacia in the periventricular white matter. A CT scan performed on the sixth day of life showed markedly reduced volume of the supratentorial brain, with diffuse and irregular thinning of the cerebral hemispheres, including both the cortex and the white matter, ex-vacuo ventricular dilatation, and simplification of the gyral pattern. There were tiny hyperdense foci adjacent to the parenchymal surface suggesting residual hemorrhage or fine calcifications. Abdominal ultrasound and echocardiogram were normal.

Epileptic seizures were detected at one month of life and were controlled with phenobarbital. An electroencephalogram performed at 3 months of age showed slowed and disorganized baseline brain electrical activity without epileptiform activity. At 69 days of life, the baby’s length (53.6 cm; −1.63 Z) and weight (5600 g; +0.81 Z) followed the expected growth for her age, yet the head circumference presented with a much slower growth (32.5 cm; −5.10 Z). At six months, she did not reach any of the development milestones. The baby’s length was 59 cm (−2.34 Z), weight was 9120 g (+2.20 Z), and head circumference cranial growth decreased even further (35 cm; −6.0 Z) (Figure 1).

**Figure 1 viruses-17-00397-f001:**
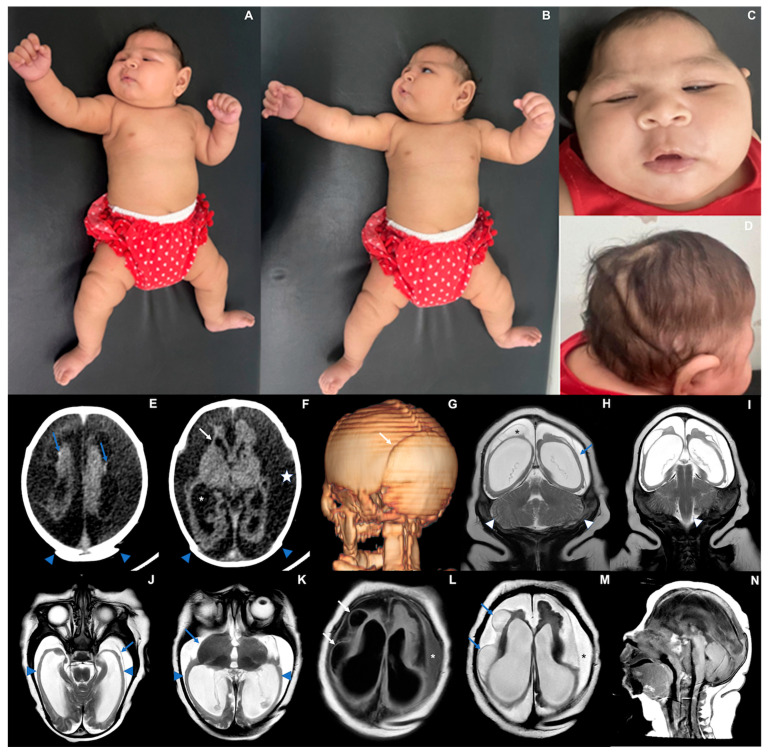
Case 1. Photographs taken at three months of life (**A**) Exacerbation of the tonic-cervical reflex and dystonia. (**B**) Retrognathia. (**C**) Small skull, head circumference 34.3 cm (−4.2 Z), epicantal folds, supratemporal depression, eyes are closed. (**D**) Redundant and folded loose skin and the appearance of a short neck, sparse and fine hair. (**E**–**G**) CT scan at six days of life showing overlapping cranial sutures ((**E**,**F**) blue arrowheads; (**G**) white arrow) and marked bilateral cerebral volume loss, with variable parenchymal thinning and irregularity ((**F**) white arrow), ventriculomegaly ((**F**) asterisk), with some tiny hyperdense foci along the cortico-pial surface (hemorrhage residues versus calcifications—(**E**) blue arrows). (**H**–**N**) MRI at three months of life demonstrating striking supratentorial parenchymal volume loss ((**H**) blue arrow) and areas of smooth cortical surface that might represent lissencephaly ((**J**,**K**) blue arrowheads). There is also secondary detachment of dura ((**J**) blue arrow) and subdural effusion ((**F**) white star; (**L**,**M**) asterisks). Cysts along the brain surface are either areas of cystic encephalomalacia and/or evidence of prior subpial hemorrhage ((**L**,**M**) arrows; (**H**) asterisk). Suspected striatal fusion is shown on (**K**) (blue arrow). The posterior fossa seems unaffected, with the brainstem and cerebellum essentially normal, allowing for image resolution ((**H**,**I**) white arrowheads, and (**N**)).

At three months of life, an MRI scan revealed extensive areas of smooth cortical surface, more evident in the temporal lobes. Some areas in the occipitotemporal transition resembled polymicrogyria, with small gyri and shallow sulci. In addition, multiple peripherally located cystic lesions throughout the juxtacortical regions of cerebral hemispheres were observed, either corresponding to cystic encephalomalacia and/or chronic sequelae of subpial hemorrhages (albeit no hemosiderin staining could be detected by T2* GRE imaging). A corpus callosum was not identified. The thalami showed volume loss, and the basal nuclei were asymmetric. Spinal cord MRI and electroneuromyography were normal. The cortical auditory evoked potentials and the fundus examination were normal at 3 months. However, the baby presented with nystagmus and visual impairment at 6 months.

### 3.2. Case 2 (Figure 2)

Case 2 is a male infant born at 36 weeks of pregnancy in June 2024. The mother reported fever, rash, headache, myalgia, and retro-orbital pain between the 8th and 9th week of pregnancy. Maternal serological tests detected OROV-IgM and ruled out other infections (Supplemental). During her 33rd week of pregnancy, after an abnormal ultrasound, a fetal MRI was performed, showing oligohydramnios, fetal hydrops, microcephaly, diffuse brain parenchymal thinning, absent corpus callosum with interhemispheric cyst, and severe supratentorial ventriculomegaly. Sequelae of a subpial hemorrhage were observed. The cerebellum and brainstem showed marked volume loss with increased cerebrospinal fluid spaces and posterior fossa subdural effusions. The baby was born alive at 36 weeks of pregnancy. His length was 45 cm (−1.06 Z), weight was 2108 g (−1.46 Z), and head circumference was 26 cm (−4.4 Z). Serological testing was positive for OROV-IgM in cerebrospinal fluid and serum. Generalized edema (fetal hydrops), marked microcephaly with craniofacial disproportion, overlapping cranial sutures, skin folds in the head, multiple arthrogryposis, camptodactyly, and cryptorchidism were observed (Table 1). The baby was extremely hypotonic, with no primitive or deep tendon reflexes, absent sucking, and very slow swallowing, and invasive ventilation was needed. X-rays showed pleural effusion and bilateral humeral bone fractures. The echocardiogram was normal. The fundus examination showed bilateral corneal edema, optic disc atrophy, and chorioretinal scar. The transfontanellar ultrasound showed laterally displaced cerebral hemispheres with undefined gyri and an abnormal posterior fossa. The transfontanellar ultrasound showed laterally displaced cerebral hemispheres with undefined gyri and an abnormal posterior fossa. A CT scan showed the same pattern of findings depicted on the fetal MRI. As the newborn had no spontaneous movements, the treatment for the humeral fracture consisted of adequate positioning, which was consolidated on his 30th day of life. This baby died at 47 days of life, and post-mortem molecular analysis through real-time RT-qPCR identified the OROV genome in several tissues, including the brain, lung, kidney, cerebrospinal fluid, and pleural fluid (Table 2 and Figure 2).

**Figure 2 viruses-17-00397-f002:**
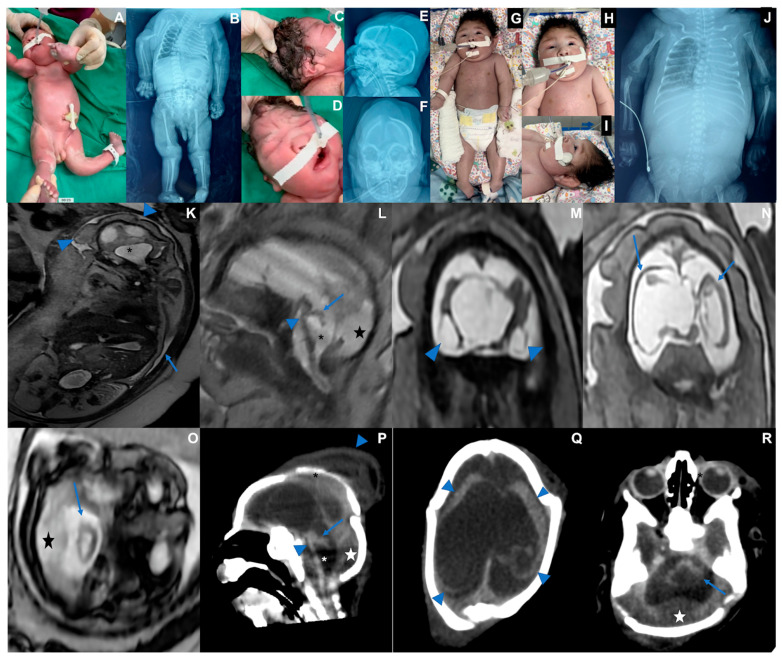
Case 2. Photographs taken at birth. (**A**) Fetal hydrops and fetal akinesia syndrome, genu recurvatum, arthrogryposis multiplex, hand camptodactyly. (**B**) X-rays showing bilateral humerus bone fractures, congenital hip dislocation, and scoliosis. (**C**) Microcephaly, skull with loose and folded skin, short neck, curly hair. (**D**) Bulging single fold, with a crease above the nose, (**E**) Overlapping cranial sutures, significant subcutaneous edema. (**F**) Craniofacial disproportion. (**G**–**J**). Thirty days old. (**G**) Multiple arthrogryposis, hyperextension of the elbows, wrists, and knees. (**H**) Spontaneous eye opening. (**I**) Cranial facial disproportion, short neck. (**J**) Consolidated bilateral humeral fracture, scoliosis, pleural effusion, bilateral hip dislocation. (**K**–**O**) Fetal MRI at the 33rd week of pregnancy showing oligohydramnios ((**K**) blue arrow) as well as microcephaly and hydrops stigmata, with redundant, folded, and diffusely thickened scalp ((**K**) blue arrowheads). Severely hypoplastic brainstem and cerebellum ((**L**,**O**,**P**,**R**) blue arrows and arrowheads), associated with an enlarged IV ventricle ((**K**,**L**,**P**) asterisk) and an increased posterior infratentorial subdural space, possibly effusion due to volume loss ((**L**,**O**,**P**,**R**) star). Bilateral cerebral mantle thinning, with a smooth outer surface ((**N**) blue arrows; (**Q**) blue arrowheads), and severe ventriculomegaly. The corpus callosum was not discerned, with a possible interhemispheric cyst. Peripherally located cystic collections appear to be present adjacent to temporal lobes, either representing cystic encephalomalacia or chronic subpial hemorrhage ((**M**) arrowheads). Cranial deformity with partially overlapping sutures and a caput succedaneum is seen on postnatal CT (**P**, downward arrowhead, **Q**).

### 3.3. Case 3 (Figure 3)

Case 3 is a female infant born at 37 weeks in July 2024. Between the 8th and 9th weeks of pregnancy, the mother reported fever, rash, headache, myalgia, retro-orbital pain, and vomiting. She tested positive for OROV-IgM and negative for other infections (Supplemental). On her 32nd week of pregnancy, a fetal ultrasound detected a fetus with microcephaly, ventriculomegaly, hypoplasia of the cerebellum, and polyhydramnios. The baby was born alive, and her length was 45 cm (−1.34 Z), her birth weight was 1.826 g (−2.57 Z), and her head circumference was 28 cm (−3.52 Z). Laboratory tests collected on the 5th day after birth were positive for OROV-IgM in cerebrospinal fluid and serum (Supplemental). The newborn presented with marked microcephaly with craniofacial disproportion, overlapping cranial sutures, multiple arthrogryposis, camptodactyly, right club foot, and subcutaneous edema. The neurological exam identified pyramidal signs with hypotonia, absence of deep tendon reflexes, lack of sucking, and very slow swallowing (Table 1). To date, she has not had any epileptic seizures. The electroencephalogram performed at 1 month of age showed slowed and disorganized baseline brain electrical activity with epileptiform activity characterized by sharp waves and multifocal polyspikes. Abdominal ultrasound showed ascites. Fundus imaging showed a bilateral chorioretinal macular scar associated with focal pigment mottling and vascular attenuation. Echocardiogram and electroneuromyography were normal.

Transfontanellar ultrasound showed ventriculomegaly, choroid plexus cysts, and posterior fossa abnormalities. A CT scan showed cerebral hemispheres with no discernible gyri and only a few sulci, including shallow Sylvian fissures. Lateral ventricles were enlarged and slightly dysmorphic. The posterior fossa was remarkably large, filled with fluid, lacking a defined cerebellum and hypoplastic corpus callosum. No calcification was detected. An MRI at 13 days of life showed a pattern of cortical development malformation characterized by nearly diffuse smooth-surfaced cerebral hemispheres with anterior-predominant agyria resembling lissencephaly, accompanied by a thin cortex. There were also areas of polymicrogyria, notably in the parietal and occipital lobes, dilated supratentorial ventricles, and increased pericerebral cerebrospinal fluid spaces. There was cystic enlargement of the posterior fossa, with wide communication between the fourth ventricle and the cisterna magna, and severe hypoplasia of the brainstem and cerebellum. Only a vestigial upwardly-rotated cerebellar vermis could be distinguished, similar to what has been described as near cerebellar agenesis in certain types of micro-lissencephaly. Traces of hypointense material were detected on inside occipital horns, suggesting prior intraventricular hemorrhage. The spinal cord MRI showed global thinning, more prominent in the thoracic segment.

On her 41st day of life, length was 47 cm (−3.38 Z), weight was 2795 g (−2.73 Z), and growth of the head circumference was 28 cm (−7.24 Z). At three months of age, her length was 50 cm (−3.81 Z), weight 3875 g (−2.29 Z), and markedly reduced head circumference growth (28.5 cm; −9.04 SD). She has torticollis and arthrogryposis of the axial and appendicular skeleton and did not reach any development milestones (Figure 3).

**Figure 3 viruses-17-00397-f003:**
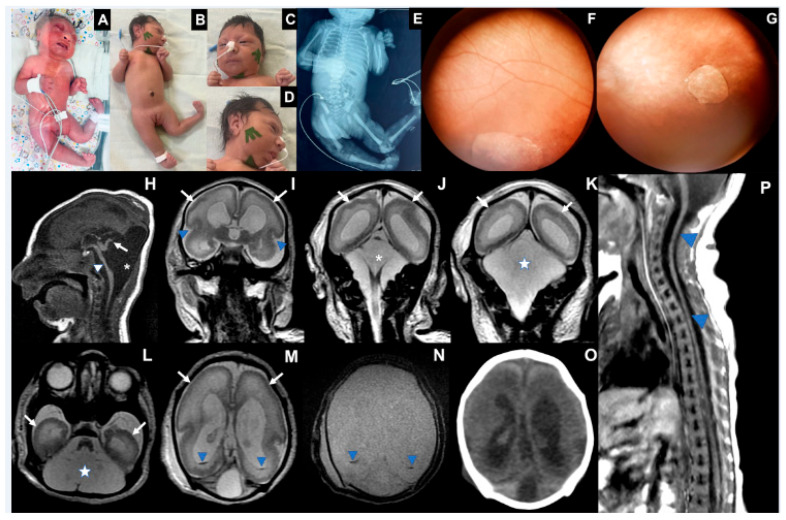
Case 3. (**A**) Photograph at birth: microcephaly, arthrogryposis multiplex, clubfeet, ankylosis, camptodactyly, subcutaneous edema. (**B**) Nineteen days of life: microcephaly, craniofacial disproportion, short neck, arthrogryposis multiplex, clubfeet, camptodactyly (**C**). Eyes are closed, epicantal folds, camptodactyly, (**D**). Microcephaly, cranial disproportion, retrognathia, reduction of edema. (**E**) X-ray with joint contractures and hip dislocation. (**F**) Right eye fundus image showing a large, well-defined chorioretinal macular scar with pigment dispersion on its margin and macular region. (**G**) Left eye fundus image showing optic disc pallor, increased disc cupping, vascular attenuation, and a well-defined chorioretinal macular scar with focal pigment mottling on its margin and macular region. (**H**–**N**) MRI at the 13th day of life: severe brainstem (**H**, arrowhead) and cerebellar hypoplasia, with an upwardly-rotated vestigial vermis (**H**, arrow); cystic dilatation of the posterior fossa (**H**,**J**, asterisk; **K**,**L**, star), communicating fourth ventricle and cisterna magna, reminiscent of Dandy–Walker malformation. Diffuse smooth-surfaced cerebral hemispheres (**I**–**M**, arrows) with shallow sylvian fissures (**I**, arrowheads) indicate lissencephaly type I pattern. Trace intraventricular hemorrhage is seen on occipital horns (**M**,**N**, blue arrowheads). No calcification was seen on the CT scan (**O**). MRI of the spine (**P**) revealed global spinal cord thinning (blue arrowheads), mainly on the thoracic segment.

## 4. Discussion

Here, we described the clinical manifestations in three live-born infants with possible congenital Oropouche infection. IgM antibodies specific to OROV before the fifth day of life in these three babies are highly suggestive of congenital infection. However, a significant limitation of our study is that we did not perform the plaque reduction neutralization test (PRNT) to eliminate the possibility of cross-reaction, especially in cases 1 and 3 where RT-PCR did not detect viral RNA [7,8]. The mother of case 1 tested positive for IgG for the Zika virus, but RT-PCR and IgM were negative. This is not an uncommon finding in a region where flaviviruses are endemic.

The three cases presented with marked microcephaly and a disproportionately small skull, collapse of the cranial vault, cranial skin folds, and a short neck, a pattern known as fetal brain disruption sequence (FBDS) [9]. The phenotype of the FBDS seen on brain imaging of these babies with OROV fetal infection is distinctive and appears to combine disruptive events such as hemorrhage as well as developmental anomalies of neurogenesis and neuronal migration. In addition to fetal viral infections and possibly extensive hemorrhage in the setting of the demise of a fetus in a twin pregnancy [10], the FBDS can result from genetic mutations. JAM3 causes AR Porencephaly-microcephaly-bilateral congenital cataract syndrome (HDBSCC, MIM #613730), in which extensive hemorrhage appears to be the main event causing the FBDS [11]. AR micro-lissencephaly caused by a mutation in NDE1 (Lissencephaly 4 with microcephaly (OMIM #609449)), a regulator of cell cycle progression, leads to the FBDS by interfering with neuronal proliferation and migration [12]. Similarly, AR microcephaly 16 (MCPH16, MIM#616681) causes the FBDS with a broad range of brain abnormalities, including simplified cortical gyral pattern, caused by mutations in ANKLE2, which also regulates neural cell division [13]. Although the OROV phenotype is not identical to the ones caused by other viral infections or these genetic mutations, the similarities suggest there could be common pathways leading to the brain structural abnormalities.

The phenotype of congenital OROV herein observed resembles the congenital Zika syndrome (CZS) in many aspects, including microcephaly, craniofacial disproportion, congenital contractures, and fundus findings such as optic disc atrophy and chorioretinal scar with focal pigment mottling (Table 2) [14]. However, the main abnormalities of the central nervous system observed in brain images have some important differences from the CZS. Infants with CZS usually had cortico-subcortical atrophy with ex-vacuo ventricular dilatation, gyral simplification with areas of polymicrogyria, subcortical and basal ganglia calcifications, increased fluid spaces, and cerebellar hypoplasia [10,14]. In the cases presented here, we observed predominantly supra- or infratentorial cortico-subcortical atrophy, ex-vacuo ventricular dilation, some features suggestive of lissencephaly–pachygyria complex, and the absence of unequivocal cerebral calcifications. There were peripheral cystic lesions along the cortical surfaces, which could either represent cystic encephalomalacia and/or chronic subpial hemorrhages (cases 1 and 2) that were not common in the images of CZS, in addition to intraventricular hemorrhage in case 3. The hypothesis of late-stage subpial hemorrhages was based on the morphology of the cystic lesions [15,16], the age of subjects (neonates), and the fact that hemorrhage can be one of the clinical consequences of OROV infection in postnatal life [17]. Particularly in case 1, since there was a considerable time interval between CT and MRI, we hypothesized the tiny hyperdense foci initially seen on CT might well represent hemorrhagic residues of SpH that were not possible to detect on T2* GRE due to either resorption and/or technical limitations of the pulse sequence [18]. Arthrogryposis and abnormal joint position possibly resulted from fetal akinesia, as observed in cases 2 and 3. Fetal akinesia is suggestive of damage to the anterior horn of the spinal cord, which was seen in the imaging of case 3. There is a difference in the clinical manifestations and severity between case 1, with less severe manifestations, and cases 2 and 3, possibly due to the different timing of exposure to the OROV.

Of the three patients, only one was diagnosed with epilepsy. In CZS, epilepsy is extremely prevalent, reaching approximately 70% of patients [19]. The actual prevalence of epilepsy in this group of congenital OROV is still biased by the small number of patients and also by age. In patients with CZS, the onset of seizures occurred mainly in the second half of the first year. The median time of the first report of seizure activity was 192 days of life [20]. Two of the present cases had disturbance of sucking and slow swallowing, also described for the CZS [21]. Thus, fundus alterations were only detected in cases 2 and 3 and were similar to those described in CZS, mainly affecting the retina and optic disc [22,23]. Hearing abnormalities are common in the CZS [24]. In the patients presented here, the Cortical Auditory Evoked Potentials investigation was normal in cases 1 and 3.

Another important aspect is the similarity with the teratogenic Orthobunyavirus infection in animals. The teratogenicity in the central nervous system of this class of viruses is well-known in ruminants. In 2011, a congenital syndrome of arthrogryposis and hydranencephaly in sheep and cattle appeared in the Netherlands due to the Schmallenberg virus, a member of the Orthobunyavirus Family [25]. The affected fetuses presented with flattened skull, cerebellar, brainstem, and spinal cord dysplasia, with a deficiency of ventral nerve roots [26]. The authors interpreted the musculoskeletal abnormalities as secondary to the brain and spinal cord damage, similar to cases 2 and 3. The hydranencephaly/arthrogryposis syndrome in ruminants has been known for decades for the Simbu serogroup, which includes the Akabane [27] and Aino víruses [28]. The Bunyamwera and the California serogroups have also shown teratogenic effects in ruminants [29,30]. The pattern is similar to what we observed here in the human babies with markedly reduced volume of the brain, diffuse brain parenchymal thinning, including both cortex and white matter, ex-vacuo ventricular dilatation and simplification of gyral pattern, severe hypoplastic brainstem and cerebellum, arthrogryposis, anasarca, and oligohydramnios. A paper published in 1995 in the USA [31] reported an association between microcephaly or macrocephaly in newborns and the presence of antibodies for different Bunyamwera serogroup viruses.

One question is why OROV has been circulating in the Americas since the 1960s without any noticeable teratogenic potential, except for suspected miscarriage cases in 1981 in Manaus [32]. Some explanations might be hypothesized: 1. The lineage presently is more pathogenic than the classic one that circulated in the Amazon region in the early decades [33,34]; 2. Since ZIKV and OROV co-circulated, some congenital cases of Oropouche might be diagnosed as congenital Zika; and 3. Limited laboratory capacity to carry out specific laboratory testing for OROV on a population level until recently [34]. These factors, and possibly others, including the absence of an approach for training medical professionals, should have missed many cases of a possible congenital disease caused by OROV in the several epidemics reported in Brazil and other endemic countries.

Future epidemiological studies are warranted to confirm the probability of vertical transmission in the different periods of pregnancy and other associated modifying factors in clinical expression. Also, additional studies to clarify the mechanism of maternal-fetal infection, such as the analysis of infected placentas, are essential [34,35].

## Figures and Tables

**Table 1 viruses-17-00397-t001:** Maternal characteristics and clinical description of the three cases with related OROV-congenital anomalies.

	Case 1	Case 2	Case 3
Maternal			
Prenatal Oropouche Fever (WG *)	21–22	8–9	8–9
Maternal age (years)	22	33	37
Drugs/Health conditions	No	No	Diabetes (from 33 WG)
Consanguinity	No	2° degree	No
Previous pregnancies	0	3	1
Positive family history	No	No	No
Birth			
Gestational age (weeks)	38	36	37
Head circumference (cm, Z-score **)	30 (−2.66)	26 (−4.40)	28
Weight (grams, Z-score)	2740 (−0.75)	2108 (−1.46)	1826
Length (cm, Z-score)	45 (−1.8)	45 (−1.06)	45
Physical findings			
Microcephaly/Craniofacial disproportion/Skull collapse/short neck	Yes	Yes	Yes
Arthrogryposis multiplex	No	Yes	Yes
Fetal hydrops	No	Yes	Yes
Neurologic findings	Hypotonia, deep tendon hyperreflexia, dystonia, and epileptic seizures	Severe hypotonia (flaccid paralysis), no primitive or deep tendon reflexes, lack of suction, slow swallowing	Hypotonia, no deep tendon reflexes, lack of suction, slow swallowing
Ophthalmological findings	Visual impairment and nystagmus at 3 months	Corneal edema, optic disc atrophy, and chorioretinal scar in OU	Optic disc pallor and increased disc cupping (OS), chorioretinal macular scar and focal pigment mottling (OU), and vascular attenuation (OS)
**Additional Exams**			
G-Banded Karyotype	NA	46, XY	46, XX
Array-SNP	NA	NA	Normal
Abdominal US ***	Normal	Pleural Effusion	Ascites
Echocardiogram	Patent oval foramen	Normal	Normal
Brainstem evoked response	Normal	NA	Normal
Electroneuromyography	Normal	NA	Normal
Brain and Spinal Cord Images	Overlapping sutures, marked cerebral atrophy, partial lissencephaly, absent corpus callosum, brain cysts (cystic encephalomalacia and/or chronic subpial hemorrhages), subdural effusion, suspected striatal fusion, normal posterior fossa, normal spinal cord	Overlapping sutures, marked cerebral atrophy, absent corpus callosum, diffuse lissencephaly, severe brainstem and cerebellar hypoplasia, brain cysts (cystic encephalomalacia and/or chronic subpial hemorrhages), subdural effusion	Overlapping sutures, moderate cerebral atrophy, hypoplastic corpus callosum, diffuse lissencephaly, severe brainstem and cerebellar hypoplasia with Dandy–Walker-like features, intraventricular hemorrhage Global thinning of the spinal cord, mainly on the thoracic segment

* WG: weeks of gestation, ** Z-score using Intergrowth 2st charts (https://intergrowth21.tghn.org/standards-tools/, accessed on 2 September 2024), *** US: ultrasound, NA: not available, OS: left eye, OU: both eyes. Array--SNP: single nucleotide polymorphism array.

**Table 2 viruses-17-00397-t002:** Laboratory tests from congenital OROV-infected cases and their mothers.

	Biological Sample	Time of Sampling (Postpartum Days)	RT-qPCR	OROV ELISA IgM	Titer (HI)	Additional Serological Data
Case 1 Infant	Serum	2	Not Detected	Positive OD 0.485	1:640	Toxo (CLIA IgG/IgM): negative Rubella, CMV, Herpes (CLIA IgG): positive Rubella, CMV, Herpes (CLIA IgM): negative DENV, CHKV, ZIKV (ELISA IgM and and RT-qPCR): negative
	Urine	7	Not Detected	NA	NA	NA
Case 1 Mother	Serum	1	Not Detected	Positive OD 1.020	≥1:1280	** HIV, Syphilis, Hepatitis B and C: negative Toxo (CLIA IgG/IgM): negative Rubella, CMV, Herpes (CLIA IgG): positive ZIKV (ELISA IgG): positive Rubella, CMV, Herpes (CLIA IgM): negative DENV, CHKV, ZIKV (ELISA IgM and RT-qPCR): negative
Case 2 Infant	Serum	1	Not Detected	Positive OD 0.725	1:80	Toxo (CLIA IgG/IgM): negative Rubella, CMV, Herpes (CLIA IgG): positive Rubella, CMV, Herpes (CLIA IgM): negative DENV, CHKV, ZIKV (ELISA IgM and RT-qPCR): negative
Post-Mortem tissues	CSF	47	Detected CT 31.7	Positive OD 1.029		DENV, CHKV, ZIKV (RT-qPCR): negative
	Pleural Fluid		Detected CT 33.7	NA		DENV, CHKV, ZIKV (RT-qPCR): negative
	Brain		Detected CT 20.4	NA		DENV, CHKV, ZIKV (RT-q-PCR): negative
	Lung		Detected CT 36.9	NA		NA
	Kidney		Detected CT 31.4	NA		NA
	Liver		Not Detected	NA		NA
	Spleen		Not Detected	NA		NA
	Heart		Inconclusive	NA		NA
Case 2 Mother	Serum	1	NA	Positive OD 0.363	1:320	** HIV, Syphilis, Hepatitis B and C: negative Toxo (CLIA IgG/IgM): negative Rubella, CMV, Herpes (CLIA IgG): positive Rubella, CMV, Herpes (CLIA IgM): negative DENV, CHKV, ZIKV (ELISA IgM): negative
Case 3 Infant	Serum	5	Not Detected	Positive OD 0.789	1:160	Toxo, Rubella, CMV, Herpes (CLIA IgG): positive Toxo, Rubella, CMV, Herpes (CLIA IgM): negative DENV, ZIKV, CHKV (ELISA IgM and RT-qPCR): negative
	CSF	19	Not Detected	Positive OD 0.653		DENV, ZIKV, CHKV (ELISA IgM): negative
Case 3 Mother	Serum	2	NA	Positive OD 0.368	1:160	** HIV, Syphilis, Hepatitis B and C: negative Toxo, CMV, Rubella (CLIA IgG): positive Toxo, CMV, Rubella (CLIA IgM): negative DENV, CHKV, ZIKV (ELISA IgM): negative

Legend: CSF, cerebrospinal fluid. NA, not available. RT-qPCR, real-time reverse quantitative transcription–polymerase chain reaction. CT, Cycle Threshold; OD. Optical Density. OROV, Oropouche virus. HIV, human immunodeficiency virus. Toxo, toxoplasmosis. CMV, cytomegalovirus. ZIKV, Zika virus. DENV, Dengue virus. CHKV, Chikungunya virus. ** Immunochromatographic rapid test was performed on the birth day. The test is for qualitative detection of Treponema Pallidum antibodies (sensitivity 99.5%, specificity 99.8%), HIV 1/2 virus-antibodies (sensitivity 99.9%, specificity 99.8%), hepatitis B surface antigen (HBsAg) (sensitivity 99.9% specificity 99.8%) and HCV antibodies (sensitivity 100%, specificity 99.7%). Toxoplasmosis, rubella, cytomegalovirus, herpes: examined by LIAISON ^®^ (Diasorin) chemiluminescent immunoassay (CLIA). (Toxoplasmosis IgG sensitivity 100% specificity 99.43%, Toxoplasmosis IgM sensitivity 100% specificity 98.49%, Rubella IgG sensitivity 99.9% specificity 99.8%, RubelIa IgM sensitivity 99.9% specificity 99.8%, Cytomegalovirus IgG sensitivity 99.5% specificity 99.74%, Cytomegalovirus IgM sensitivity 90.67% specificity 99.07%, Herpes IgG sensitivity 99.42% specificity 96.84%, Herpes IgM sensitivity 100% specificity 96.45%). ZIKV, DENV, CHKV: tested at the Instituto Evandro Chagas by RT-qPCR and enzyme-linked immunosorbent (ELISA IgM) (in-house), which has a sensitivity of 93% and a specificity of 99%. OROV: tested at the Instituto Evandro Chagas by RT-qPCR and ELISA IgM (in-house), which has a sensitivity of 93% and a specificity of 99% and hemagglutination inhibition (HI).

## Data Availability

Additional data on patients’ laboratory results or clinical details can be obtained from the corresponding author (L.S.-F.).

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
