# Peer review of "Congenital Oropouche in Humans: Clinical Characterization of a Possible New Teratogenic Syndrome"

_viruses, 2025, doi:10.3390/v17030397_

Round 1
Reviewer 1 Report
Comments and Suggestions for Authors
The authors present three case studies of infants with evidence of congenital Oropouche virus infection. The infection appears well documented in all three infants and the case presentations are engaging and complete. Only one minor consideration for Case 1 is the detection of Zika virus IgG antibody in mom, which of course can happen in a Zika virus endemic area from a past or recent infection. The only Zika virus test done on the infant was a Zika IgM. Zika virus detection/diagnostic methods are notoriously insensitive, and elusive. Can the authors provide any ZIka PCR data on urine, serum or CSF for Case 1? Or at least comment on line 156, the minor limitations of not fully excluding congenital Zika virus infection in Case 1.
Also, any retinal photographs available as the authors describe a retinitis in the infants.
Schwartz et al Viruses 2024 Sep 9, 16(9):1435 also has a nice review published, the author smay wish to include in references/citations as background of current knowledge.
Author Response
COMMEnt 1: The authors present three case studies of infants with evidence of congenital Oropouche virus infection. The infection appears well documented in all three infants and the case presentations are engaging and complete. Only one minor consideration for Case 1 is the detection of Zika virus IgG antibody in mom, which of course can happen in a Zika virus endemic area from a past or recent infection. The only Zika virus test done on the infant was a Zika IgM. Zika virus detection/diagnostic methods are notoriously insensitive, and elusive. Can the authors provide any ZIka PCR data on urine, serum or CSF for Case 1? Or at least comment on line 156, the minor limitations of not fully excluding congenital Zika virus infection in Case 1.
RESPONSE 1: Thank you for noting this. We noticed that some information in the Table 2 was missing. We tested serum RT-PCR for Zika virus from Case 1. We added this information on Table 1 and other relevant tests from this and other cases. We also added the optical densities for IgM and the cycle threshold for RT-PCR in Table 2. We also reformulated the discussion to include this point as follows:
IgM antibodies specific to OROV before the fifth day of life in these three babies are highly suggestive of congenital infection. However, a significant limitation of our study is that we didn’t perform the Plaque Reduction Neutralization Test (PRNT) to eliminate the possibility of cross-reaction, especially in cases 1 and 3 where RT-PCR didn’t detect viral RNA [7,8]. The mother of case 1 tested positive for IgG for Zika Virus, but RT-PCR and IgM were negative. This is not an uncommon finding in a region where flaviviruses are endemic.
COMMENT 2: Also, any retinal photographs available as the authors describe a retinitis in the infants.
RESPONSE 2: Retinal images from Case 1 were not available due to technical difficulties since the baby had a continuous eye movement. For Case 2, unfortunately the baby died before we could obtain retinal images. Case 3 has images recorded on Figure 3.
Comment 3: Schwartz et al Viruses 2024 Sep 9, 16(9):1435 also has a nice review published, the authors may wish to include in references/citations as background of current knowledge.
RESPONSE 3: Thank you. Very important reference and we added to our discussion. Also, there is a more recent publication from Schwartz (2025) we decided to include both publications. We also improved our discussion with information from this publication
- Schwartz DA. Novel Reassortants of Oropouche Virus (OROV) Are Causing Maternal-Fetal Infection During Pregnancy, Stillbirth, Congenital Microcephaly and Malformation Syndromes. Genes (Basel). 2025 Jan 15;16(1):87. doi: 10.3390/genes16010087.
- Schwartz DA, Dashraath P, Baud D. Oropouche Virus (OROV) in Pregnancy: An Emerging Cause of Placental and Fetal Infection Associated with Stillbirth and Microcephaly following Vertical Transmission. Viruses. 2024 Sep 9;16(9):1435. doi: 10.3390/v16091435.
Reviewer 2 Report
Comments and Suggestions for Authors
1. Table: "NDet" should be changed to "Not Detected" ; "Det" should be changed to "Detected"
2. Fig. 1-3: covering the eyes with a black bar is a common practice to protect a patient's privacy by obscuring their identity.
3. A recent infection can be confirmed by testing acute and convalescent samples using serologic testing to look for a four‐fold or greater change in antibody titers. Did the authors look at the titer change?
4. Typo in line 385: "Since ZIKV e OROV co-circulated" should be " ZIKV and OROV "
5. Discussion: Line 381-382 "limited laboratory capacity to carry out specific laboratory testing for OROV" cannot fully explain "why OROV has been circulating in the Americas since the 1960s without any noticeable teratogenic potential".
6. Discussion: Line 305-307: "Unlike the flavivirus, where cross-reaction is common, the OROV IgM and IgG antibodies are highly specific [8]." ref8 has nothing to do with OROV. Lack of PRNT is a limitation of this study. PRNT is recommended as cross-reaction may occur for serology testing. More references should be cited, with this one (PMID: 39221481) as an example.
Author Response
COMMENT 1. Table: "NDet" should be changed to "Not Detected" ; "Det" should be changed to "Detected"
RESPONSE 1: Thank you. We changed it following your suggestion. We also noticed that some information in the Table 2 was missing. We tested serum RT-PCR from Case 1. We added this information on Table 1, as well as other relevant tests from this and other cases, such as array-CGH in addition to the Karyotype.
COMMENT 2. Fig. 1-3: covering the eyes with a black bar is a common practice to protect a patient's privacy by obscuring their identity.
RESPONSE 2 We discussed with the three families and they agreed to not cover the eyes since we believe that the full face is important for the phenotype recognition. In their consent form, the guardians acknowledged that photos with the faces of the children would be published in journal of open access. We clarified this point in the Methods section as follows:
The mothers filled out the informed consent form, including the publication of photographs of the full face.
COMMENT 3. A recent infection can be confirmed by testing acute and convalescent samples using serologic testing to look for a four‐fold or greater change in antibody titers. Did the authors look at the titer change?
RESPONSE 3. We didn’t look for IgG change in antibody titers since at the time the mothers had their diagnosis, Oropouche was not a concern for a congenital syndrome. We only assessed their serum results after their babies were born with microcephaly. We considered the IgM positivity a proxy for acute infection associated with symptoms compatible with OROV infection. But you are right. Prospective pregnant women should have both acute and convalescent serum tested. We added more information on Table 2 including RT-PCR for other viruses in the mother (including ZIKV) and the optical densities for IgM and the cycle threshold for RT-PCR. We added the results of array-CGH in Table 1.
COMMENT 4. Typo in line 385: "Since ZIKV e OROV co-circulated" should be " ZIKV and OROV "
RESPONSE 4: Thank you. Corrected
COMMENT 5. Discussion: Line 381-382 "limited laboratory capacity to carry out specific laboratory testing for OROV" cannot fully explain "why OROV has been circulating in the Americas since the 1960s without any noticeable teratogenic potential".
RESPONSE 5. What we wanted to address with this sentence is that before 2016, testing for OROV in our population was not routinely used. Immunological testing for OROV is still based on in-house and time-consuming tests in Brazil, and the RT-PCR was unavailable before the present century. Moreover, before the epidemic of microcephaly associated with maternal Zika virus infection, there was no real awareness of the teratogenic potential of arboviruses. Therefore, before 2016, it would be challenging to have tests confirming or even suspecting OROV in mothers of children with microcephaly.
We modified this line in the discussion as follows: Limited laboratory capacity to carry out specific laboratory testing for OROV on a population level until recently [34].
COMMENT 6. Discussion: Line 305-307: "Unlike the flavivirus, where cross-reaction is common, the OROV IgM and IgG antibodies are highly specific [8]." ref8 has nothing to do with OROV. Lack of PRNT is a limitation of this study. PRNT is recommended as cross-reaction may occur for serology testing. More references should be cited, with this one (PMID: 39221481) as an example.
RESPONSE 6. Thank you for noticing this. We corrected the reference and added your suggestion. You are correct in that we didn’t perform PRNT, which is a limitation in this study. We added the optical densities for IgM and Cycle Threshold for RT-PCR in Table 2 (this was missing before)
The presence of IgM antibodies specific for OROV before the fifth day of life in these three babies is highly suggestive of congenital infection. However, a significant limitation of our study is that we didn’t perform the Plaque Reduction Neutralizing Antibody Test to eliminate the possibility of cross-reaction, especially in cases 1 and 3 where RT-PCR didn’t detect viral RNA [7,8].
7. Liu BM. Epidemiological and clinical overview of the 2024 Oropouche virus disease outbreaks, an emerging/re-emerging neurotropic arboviral disease and global public health threat. J Med Virol. 2024 Sep;96(9):e29897. doi: 10.1002/jmv.29897.
8. Scachetti GC, Forato J, Claro IM, et al. Re-emergence of Oropouche virus between 2023 and 2024 in Brazil: an observational epidemiological study. Lancet Infect Dis. 2025 Feb; 25(2):166-175. doi: 10.1016/S1473-3099(24)00619-4.